# NETWORK AUGMENTATION FOR TINY DEEP LEARNING

**Han Cai**[1]**, Chuang Gan**[2]**, Ji Lin**[1]**, Song Han**[1]
[1]Massachusetts Institute of Technology,     [2]MIT-IBM Watson AI Lab
https://tinyml.mit.edu

## ABSTRACT

We introduce *Network Augmentation* (NetAug), a new training method for improving the performance of tiny neural networks. Existing regularization techniques (e.g., data augmentation, dropout) have shown much success on large neural networks by adding noise to overcome over-fitting. However, we found these techniques hurt the performance of tiny neural networks. We argue that training tiny models are different from large models: rather than augmenting the data, we should augment the model, since tiny models tend to suffer from under-fitting rather than over-fitting due to limited capacity. To alleviate this issue, NetAug augments the network (reverse dropout) instead of inserting noise into the dataset or the network. It puts the tiny model into larger models and encourages it to work as a sub-model of larger models to get extra supervision, in addition to functioning as an independent model. At test time, only the tiny model is used for inference, incurring zero inference overhead. We demonstrate the effectiveness of NetAug on image classification and object detection. NetAug consistently improves the performance of tiny models, achieving up to 2.2% accuracy improvement on ImageNet. On object detection, achieving the same level of performance, NetAug requires 41% fewer MACs on Pascal VOC and 38% fewer MACs on COCO than the baseline.

## 1 INTRODUCTION

Tiny IoT devices are witnessing rapid growth, reaching 75.44 billion by 2025 (iot). Deploying deep neural networks directly on these tiny edge devices without the need for a connection to the cloud brings better privacy and lowers the cost. However, tiny edge devices are highly resource-constrained compared to cloud devices (e.g., GPU). For example, a microcontroller unit (e.g., STM32F746) typically only has 320KB of memory (Lin et al., 2020), which is 50,000x smaller than the memory of a GPU. Given such strict constraints, neural networks must be extremely small to run efficiently on these tiny edge devices. Thus, improving the performance of tiny neural networks (e.g., MCUNet (Lin et al., 2020)) has become a fundamental challenge for tiny deep learning.

Conventional approaches to improve the performance of deep neural networks rely on regularization techniques to alleviate over-fitting, including data augmentation methods (e.g., AutoAugment (Cubuk et al., 2019), Mixup (Zhang et al., 2018a)), dropout methods (e.g., Dropout (Srivastava et al., 2014), DropBlock (Ghiasi et al., 2018)), and so on. Unfortunately, this common approach does not apply to tiny neural networks. Figure 1 (left) shows the ImageNet (Deng et al., 2009) accuracy of state-of-the-art regularization techniques on ResNet50 (He et al., 2016) and MobileNetV2-Tiny (Lin et al., 2020). These regularization techniques significantly improve the ImageNet accuracy of ResNet50, but unfortunately, they hurt the ImageNet accuracy for MobileNetV2-Tiny, which is 174x smaller. We argue that *training tiny neural networks is fundamentally different from training large neural networks.* Rather than augmenting the dataset, we should augment the network. Large neural networks tend to over-fit the training data, and data augmentation techniques can alleviate the over-fitting issue. However, tiny neural networks tend to under-fit the training data due to limited capacity (174x smaller); applying regularization techniques to tiny neural networks will worsen the under-fitting issue and degrade the performance.

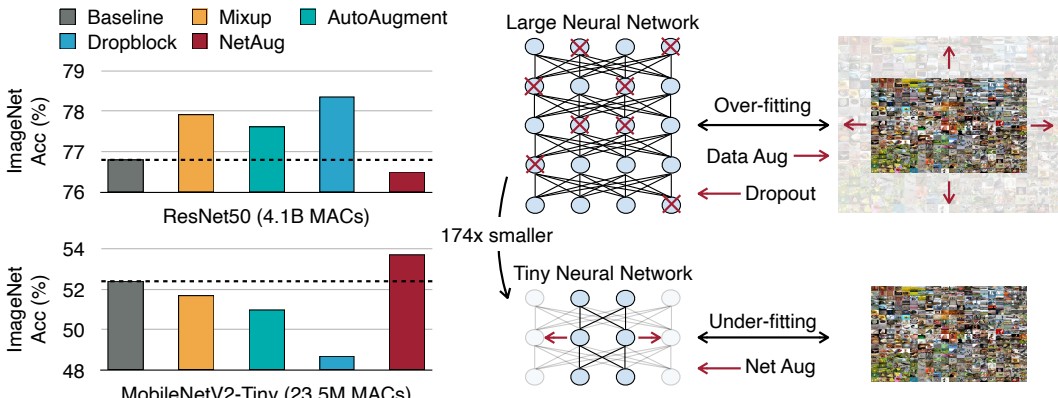

Figure 1: *Left:* ResNet50 (large neural network) benefits from regularization techniques, while MobileNetV2-Tiny (tiny neural network) losses accuracy by these regularizations. *Right:* Large neural networks suffer from over-fitting, thus require regularization such as data augmentation and dropout. In contrast, tiny neural networks tend to under-fit the dataset, thus requires more capacity during training. NetAug augments the network (reverse dropout) during training to provide more supervision for tiny neural networks. Contrary to regularization techniques, it improves the accuracy of tiny neural networks and as expected, hurts the accuracy of non-tiny neural networks.

In this paper, we propose *Network Augmentation* (NetAug), a new training technique for tiny deep learning. Our intuition is that tiny neural networks need more capacity rather than noise at the training time. Thus, instead of adding noise to the dataset via data augmentation or adding noise to the model via dropout (Figure 1 upper right), NetAug augments the tiny model by inserting it into larger models, sharing the weights and gradients; the tiny model becomes a sub-model of the larger models apart from working independently (Figure 1 lower right). It can be viewed as a reversed form of dropout, as we enlarge the target model instead of shrinking it. At the training time, NetAug adds the gradients from larger models as extra training supervision for the tiny model. At test time, only the tiny model is used for inference, causing zero overhead.

Extensive experiments on ImageNet (ImageNet, ImageNet-21k-P) and five fine-grained image classification datasets (Food101, Flowers102, Cars, Cub200, and Pets) show that NetAug is much more effective than regularization techniques for tiny neural networks. Applying NetAug to MobileNetV2-Tiny improves the ImageNet accuracy by 1.6% while adding only 16.7% training cost overhead and zero inference overhead. On object detection datasets, NetAug improves the AP50 of YoloV3 (Redmon & Farhadi, 2018) with Mbv3 w0.35 as the backbone by 3.36% on Pascal VOC and by 1.8% on COCO.

## 2 RELATED WORK

**Knowledge Distillation.** Knowledge distillation (KD) (Hinton et al., 2015; Furlanello et al., 2018; Yuan et al., 2020; Beyer et al., 2021; Shen et al., 2021; Yun et al., 2021) is proposed to transfer the "dark knowledge" learned in a large teacher model to a small student model. It trains the student model to match the teacher model's output logits (Hinton et al., 2015) or intermediate activations (Romero et al., 2015; Zagoruyko & Komodakis, 2017) for better performances. Apart from being used alone, KD can be combined with other methods to improve the performance, such as (Zhou et al., 2020) that combines layer-wise KD and network pruning.

Unlike KD, our method aims to improve the performances of neural networks from a different perspective, i.e., tackling the under-fitting issue of tiny neural networks. Technically, our method does not require the target model to mimic a teacher model. Instead, we train the target model to work as a sub-model of a set of larger models, built by augmenting the width of the target model, to get extra training supervision. Since the underlying mechanism of our method is fundamentally different from KD's. Our method is complementary to the use of KD and can be combined to boost performance (Table 2).

**Regularization Methods.** Regularization methods typically can be categorized into data augmentation families and dropout families. Data augmentation families add noise to the dataset by applying specially-designed transformations on the input, such as Cutout (DeVries & Taylor, 2017) and Mixup (Zhang et al., 2018a). Additionally, AutoML has been employed to search for a combination of transformations for data augmentation, such as AutoAugment (Cubuk et al., 2019) and RandAugment (Cubuk et al., 2020).

Instead of injecting noise into the dataset, dropout families add noise to the network to overcome overfitting. A typical example is Dropout (Srivastava et al., 2014) that randomly drops connections of the neural network. Inspired by Dropout, many follow-up extensions propose structured forms of dropout for better performance, such as StochasticDepth (Huang et al., 2016), SpatialDropout (Tompson et al., 2015), and DropBlock (Ghiasi et al., 2018). In addition, some regularization techniques combine dropout with other methods to improve the performance, such as Self-distillation (Zhang et al., 2019a) that combines knowledge distillation and depth dropping, and GradAug (Yang et al., 2020) that combines data augmentation and channel dropping.

Unlike these regularization methods, our method targets improving the performance of tiny neural networks that suffer from under-fitting by augmenting the width of the neural network instead of shrinking it via random dropping. It is a reversed form of dropout. Our experiments show that NetAug is more effective than regularization methods on tiny neural networks (Table 3).

**Tiny Deep Learning.** Improving the inference efficiency of neural networks is very important in tiny deep learning. One commonly used approach is to compress existing neural networks by pruning (Han et al., 2015; He et al., 2017; Liu et al., 2017) and quantization (Han et al., 2016; Zhu et al., 2017; Rastegari et al., 2016). Another widely adopted approach is to design efficient neural network architectures (Iandola et al., 2016; Sandler et al., 2018; Zhang et al., 2018b). In addition to manually designed compression strategies and neural network architectures, AutoML techniques recently gain popularity in tiny deep learning, including auto model compression (Cai et al., 2019a; Yu & Huang, 2019) and auto neural network architecture design (Tan et al., 2019; Cai et al., 2019b; Wu et al., 2019). Unlike these techniques, our method focuses on improving the accuracy of tiny neural networks without changing the model architecture . Combining these techniques with our method leads to better performances in our experiments (Table 1).

## 3 NETWORK AUGMENTATION

In this section, we first describe the formulation of NetAug. Then we introduce practical implementations. Lastly, we discuss the overhead of NetAug during training (16.7%) and test (zero).

### 3.1 FORMULATION

We denote the weights of the tiny neural network as $W_t$ and the loss function as $\mathcal{L}$. During training, $W_t$ is optimized to minimize $\mathcal{L}$ with gradient updates: $W_t^{n+1} = W_t^n - \eta \frac{\partial \mathcal{L}(W_t^n)}{\partial W_t^n}$, where $\eta$ is the learning rate, and we assume using standard stochastic gradient descent for simplicity. Since the capacity of the tiny neural network is limited, it is more likely to get stuck in local minimums than large neural networks, leading to worse training and test performances.

We aim to tackle this challenge by introducing additional supervision to assist the training of the tiny neural network. Contrary to dropout methods that encourage subsets of the neural network to produce predictions, NetAug encourages the tiny neural network to work as a sub-model of a set of larger models constructed by augmenting the width of the tiny model (Figure 2 left). The augmented loss function $\mathcal{L}_{\text{aug}}$ is:

$$\mathcal{L}_{\text{aug}} = \underbrace{\mathcal{L}(W_t)}_{\text{base supervision}} + \underbrace{\alpha_1 \mathcal{L}([W_t, W_1]) + \cdots + \alpha_i \mathcal{L}([W_t, W_i]) + \cdots}_{\text{auxiliary supervision, working as a sub-model of augmented models}} , \quad (1)$$

where $[W_t, W_i]$ represents an augmented model that contains the tiny neural network $W_t$ and new weights $W_i$. $\alpha_i$ is the scaling hyper-parameter for combining loss from different augmented models.

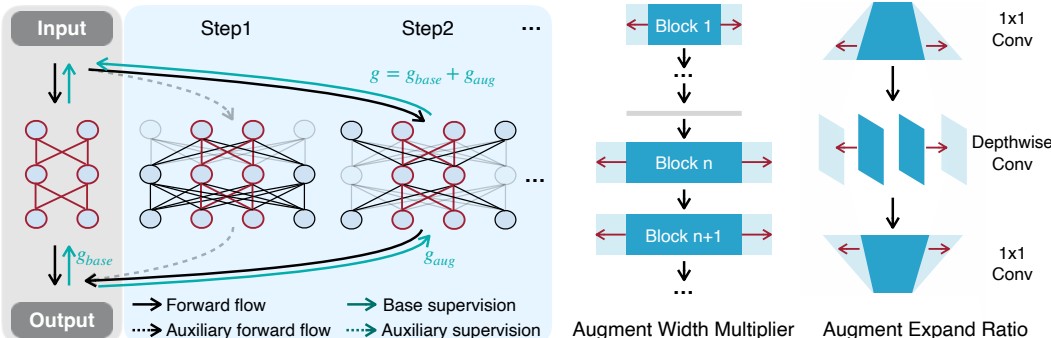

Figure 2: *Left:* We augment a tiny network by putting it into larger neural networks. They share the weights. The tiny neural network is supervised to produce useful representations for larger neural networks beyond functioning independently. At each training step, we sample one augmented network to provide auxiliary supervision that is added to the base supervision. At test time, only the tiny network is used for inference, which has zero overhead. *Right:* NetAug is implemented by augmenting the width multiplier and expand ratio of the tiny network.

## 3.2 IMPLEMENTATION

**Constructing Augmented Models.**    Keeping the weights of each augmented model (e.g., $[W_t, W_i]$) independent is resource-prohibitive, as the model size grows linearly as the number of augmented models increases. Therefore, we share the weights of different augmented models, and only maintain the largest augmented model. We construct other augmented models by selecting the sub-networks from the largest augmented model (Figure 2 left).

This weight-sharing strategy is also used in one-shot neural architecture search (NAS) (Guo et al., 2020; Cai et al., 2020a) and multi-task learning (Ruder, 2017). Our objective and training process are completely different from theirs: i) one-shot NAS trains a weight-sharing super-net that supports all possible sub-networks. Its goal is to provide efficient performance estimation in NAS. In contrast, NetAug focuses on improving the performance of a tiny neural network by utilizing auxiliary supervision from augmented models. In addition, NetAug can be applied to NAS-designed neural networks for better performances (Table 1). ii) Multi-task learning aims to transfer knowledge across different tasks via weight sharing. In contrast, NetAug transmits auxiliary supervision on a single task, from augmented models to the tiny model.

Specifically, we construct the largest augmented model by augmenting the width (Figure 2 right), which incurs smaller training time overhead on GPUs than augmenting the depth (Radosavovic et al., 2020). For example, assume the width of a convolution operation is $w$, we augment its width by an *augmentation factor* $r$. Then the width of the largest augmented convolution operation is $r \times w$. For simplicity, we use a single hyper-parameter to control the *augmentation factor* for all operators in the network.

After building the largest augmented model, we construct other augmented models by selecting a subset of channels from the largest augmented model. We use a hyper-parameter $s$, named *diversity factor*, to control the number of augmented model configurations. We set the augmented widths to be linearly spaced between $w$ and $r \times w$. For instance, with $r = 3$ and $s = 2$, the possible widths are $[w, 2w, 3w]$. Different layers can use different augmentation ratios. In this way, we get diverse augmented models from the largest augmented model, each containing the target neural network.

**Training Process.**    As shown in Eq. 1, getting supervision from one augmented network requires an additional forward and backward process. It is computationally expensive to involve all augmented networks in one training step. To address this challenge, we only sample one augmented network at each step. The tiny neural network is updated by merging the base supervision (i.e., $\frac{\partial \mathcal{L}(W_t^n)}{\partial W_t^n}$) and the auxiliary supervision from this sampled augmented network (Figure 2 left):

$$W_t^{n+1} = W_t^n - \eta \left( \frac{\partial \mathcal{L}(W_t^n)}{\partial W_t^n} + \alpha \frac{\partial \mathcal{L}([W_t^n, W_i^n])}{\partial W_t^n} \right), \tag{2}$$

where $[W_t^n, W_i^n]$ represents the sampled augmented network at this training step. For simplicity, we use the same scaling hyper-parameter $\alpha$ ($\alpha = 1.0$ in our experiments) for all augmented networks. In addition, $W_i$ is also updated via gradient descent in this training step. It is possible to sample more augmented networks in one training step. However, in our experiments, we found it not only increases the training cost but also hurts the performance. Thus, we only sample one augmented network in each training step.

## 3.3 TRAINING AND INFERENCE OVERHEAD

NetAug is only applied at the training time. At inference time, we only keep the tiny neural network. Therefore, the inference overhead of NetAug is zero. In addition, as NetAug does not change the network architecture, it does not require special support from the software system or hardware, making it easier to deploy in practice.

Regarding the training overhead, applying NetAug adds an extra forward and backward process in each training step, which seems to double the training cost. However, in our experiments, the training time is only 16.7% longer (245 GPU hours v.s. 210 GPU hours, shown in Table 3). It is because the total training cost of a tiny neural network is dominated by data loading and communication cost, not the forward and backward computation, since the model is very small. Therefore, the overall training time overhead of NetAug is only 16.7%. Apart from the training cost, applying NetAug will increase the peak training memory footprint. Since we focus on training tiny neural networks whose peak training memory footprint is much smaller than large neural networks', in practice, the slightly increased training memory footprint can still fit in GPUs.

## 4 EXPERIMENTS

### 4.1 SETUP

**Datasets.** We conducted experiments on seven image classification datasets, including ImageNet (Deng et al., 2009), ImageNet-21K-P (winter21 version) (Ridnik et al., 2021), Food101 (Bossard et al., 2014), Flowers102 (Nilsback & Zisserman, 2008), Cars (Krause et al., 2013), Cub200 (Wah et al., 2011), and Pets (Parkhi et al., 2012). In addition to image classification, we also evaluated our method on Pascal VOC object detection (Everingham et al., 2010) and COCO object detection (Lin et al., 2014)[1].

**Training Details.** For ImageNet experiments, we train models with batch size 2048 using 16 GPUs. We use the SGD optimizer with Nesterov momentum 0.9 and weight decay 4e-5. By default, the models are trained for 150 epochs on ImageNet and 20 epochs on ImageNet-21K-P, except stated explicitly. The initial learning rate is 0.4 and gradually decreases to 0 following the cosine schedule. Label smoothing is used with a factor of 0.1 on ImageNet.

For experiments on fine-grained image classification datasets (Food101, Flowers102, Cars, Cub200, and Pets), we train models with batch size 256 using 4 GPUs. We use ImageNet-pretrained weights to initialize the models and finetune the models for 50 epochs.

For Pascal VOC object detection, we train models for 200 epochs with batch size 64 using 8 GPUs. The training set consists of Pascal VOC 2007 trainval set and Pascal VOC 2012 trainval set, while Pascal VOC 2007 test set is used for testing. For COCO object detection, we train models for 120 epochs with batch size 128 using 16 GPUs. COCO2017 *train* is used for training while COCO2017 *val* is used for testing.

We use the YoloV3 (Redmon & Farhadi, 2018) detection framework and replace the backbone with tiny neural networks. We also replace normal convolution operations with depthwise convolution operations in the head of YoloV3. ImageNet-pretrained weights are used to initialize the backbone while the detection head is initialized randomly.

### 4.2 RESULTS ON IMAGENET

**Main Results.** We apply NetAug to commonly used tiny neural network architectures in TinyML (Lin et al., 2020; Saha et al., 2020; Banbury et al., 2021), including MobileNetV2-Tiny (Lin et al.,

---

[1]Code and pre-trained weights: https://github.com/mit-han-lab/tinyml

| Model | MobileNetV2 -Tiny, r144 | MCUNet r176 | MobileNetV3 r160, w0.35 | ProxylessNAS r160 w0.35 | w1.0 | MobileNetV2 r160 w0.35 | w1.0 | ResNet50 r224 |
|---|---|---|---|---|---|---|---|---|
| Params | 0.75M | 0.74M | 2.2M | 1.8M | 4.1M | 1.7M | 3.5M | 25.5M |
| MACs | 23.5M | 81.8M | 19.6M | 35.7M | 164.1M | 30.9M | 154.1M | 4.1G |
| Baseline | 51.7% | 61.5% | 58.1% | 59.1% | 71.2% | 56.3% | 69.7% | 76.8% |
| NetAug | 53.3% | 62.7% | 60.3% | 60.8% | 71.9% | 57.8% | 70.6% | 76.5% |
| ΔAcc | **+1.6%** | **+1.2%** | **+2.2%** | **+1.7%** | **+0.7%** | **+1.5%** | **+0.9%** | -0.3% |

Table 1: NetAug consistently improves the ImageNet accuracy for popular tiny neural networks. The smaller the model, the larger the improvement. 'w' represents the width multiplier and 'r' represents the input image size.

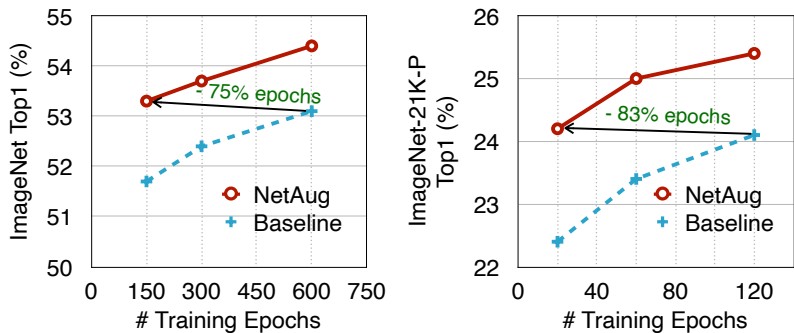

Figure 3: NetAug outperforms the baseline under different numbers of training epochs on ImageNet and ImageNet-21K-P for MobileNetV2-Tiny. With similar accuracy, NetAug requires 75% fewer training epochs on ImageNet and 83% fewer training epochs on ImageNet-21K-P.

2020), ProxylessNAS-Mobile (Cai et al., 2019b), MCUNet (Lin et al., 2020), MobileNetV3 (Howard et al., 2019), and MobileNetV2 (Sandler et al., 2018). As shown in Table 1, NetAug provides consistent accuracy improvements over the baselines on different neural architectures. Specifically, for ProxylessNAS-Mobile, MCUNet and MobileNetV3, whose architectures are optimized using NAS, NetAug still provides significant accuracy improvements (1.7% for ProxylessNAS-Mobile, 1.2% for MCUNet, and 2.2% for MobileNetV3). In addition, we find NetAug tends to provide higher accuracy improvement on smaller neural networks (+0.9% on MobileNetV2 w1.0 → +1.5% on MobileNetV2 w0.35). We conjecture that smaller neural networks have lower capacity, thus suffer more from the under-fitting issue and benefits more from NetAug. Unsurprisingly, NetAug hurts the accuracy of non-tiny neural network (ResNet50), which already has enough model capacity on ImageNet.

Figure 3 summarizes the results of MobileNetV2-Tiny on ImageNet and ImageNet-21K-P under different numbers of training epochs. NetAug provides consistent accuracy improvements over the baseline under all settings. In addition, to achieve similar accuracy, NetAug requires much fewer training epochs than the baseline (75% fewer epochs on ImageNet, 83% fewer epochs on ImageNet-21K-P), which can save the training cost and reduce the $CO_2$ emissions.

**Comparison with KD.** We compare the ImageNet performances of NetAug and knowledge distillation (KD) on MobileNetV2-Tiny (Lin et al., 2020), MobileNetV2, MobileNetV3, and ProxylessNAS. All models use the same teacher model (Assemble-ResNet50 (Lee et al., 2020)) when training with KD.

The results are summarized in Table 2. Compared with KD, NetAug provides slightly higher ImageNet accuracy improvements: +0.5% on MobileNetV2-Tiny, +0.8% on MobileNetV2 (w0.35, r160), +0.5% on MobileNetV3 (w0.35, r160), and +0.4% on ProxylessNAS (w0.35, r160). In addition, we find NetAug's improvement is orthogonal to KD's. Combining NetAug and KD can further boost the ImageNet accuracy of tiny neural networks: 2.7% on MobileNetV2-Tiny, 2.9%

| Model | Baseline | KD | NetAug | NetAug + KD |
|---|---|---|---|---|
| MobileNetV2-Tiny | 51.7% | 52.8% (+1.1%) | 53.3% (+1.6%) | **54.4% (+2.7%)** |
| MobileNetV2 w0.35, r160 | 56.3% | 57.0% (+0.7%) | 57.8% (+1.5%) | **59.2% (+2.9%)** |
| MobileNetV3 w0.35, r160 | 58.1% | 59.8% (+1.7%) | 60.3% (+2.2%) | **61.5% (+3.4%)** |
| ProxylessNAS w0.35, r160 | 59.1% | 60.4% (+1.3%) | 60.8% (+1.7%) | **61.5% (+2.4%)** |

Table 2: Comparison with KD (Hinton et al., 2015) on ImageNet. NetAug is orthogonal to KD. Combining NetAug with KD further boosts the ImageNet accuracy of tiny neural networks.

| Model (MobileNetV2-Tiny) | #Epochs | Training Cost (GPU Hours) | ImageNet Top1 Acc | $\Delta$Acc |
|---|---|---|---|---|
| Baseline | 150 | 210 | 51.7% | - |
| Dropout (kp=0.9) (Srivastava et al., 2014) | 150 | 210 | 51.0% | -0.7% |
| Dropout (kp=0.8) (Srivastava et al., 2014) | 150 | 210 | 50.3% | -1.4% |
| Baseline | 300 | 420 | 52.4% | - |
| Mixup (Zhang et al., 2018a) | 300 | 420 | 51.7% | -0.7% |
| AutoAugment (Cubuk et al., 2019) | 300 | 440 | 51.0% | -1.4% |
| RandAugment (Cubuk et al., 2020) | 300 | 440 | 49.6% | -2.8% |
| DropBlock (Ghiasi et al., 2018) | 300 | 420 | 48.7% | -3.7% |
| NetAug | 300 | 490 | **53.7%** | +1.3% |

Table 3: Regularization techniques hurt the accuracy for MobileNetV2-Tiny, while NetAug provides 1.3% top1 accuracy improvement with only 16.7% training cost overhead.

on MobileNetV2 (w0.35, r160), 3.4% on MobileNetV3 (w0.35, r160), and 2.4% on ProxylessNAS (w0.35, r160).

**Comparison with Regularization Methods.** Regularization techniques hurt the performance of tiny neural network (Table 3), even when the regularization strength is very weak (e.g., dropout with keep probability 0.9). This is due to tiny networks has very limited model capacity. When adding stronger regularization (e.g., RandAugment, Dropblock), the accuracy loss gets larger (up to 3.7% accuracy loss). Additionally, we notice that Mixup, Dropblock, and RandAugment provide hyper-parameters to adjust the strength of regularization. We further studied these methods under different regularization strengths. The results are reported in the appendix (Table 6) due to the space limit. Similar to observations in Table 3, we consistently find that: i) adding these regularization methods hurts the accuracy of the tiny neural network. ii) Increasing the regularization strength leads to a higher accuracy loss.

Based on these results, we conjecture that tiny neural networks suffer from the under-fitting issue rather than the over-fitting issue. Applying regularization techniques designed to overcome over-fitting will exacerbate the under-fitting issue of tiny neural networks, thereby leading to accuracy loss. In contrast, applying NetAug improves the ImageNet accuracy of MobileNetV2-Tiny by 1.3%, with zero inference overhead and only 16.7% training cost overhead. NetAug is more effective than regularization techniques in tiny deep learning.

**Discussion.** NetAug improves the accuracy of tiny neural networks by alleviating under-fitting. However, for larger neural networks that do not suffer from under-fitting, applying NetAug may, on the contrary, exacerbate over-fitting, leading to degraded validation accuracy (as shown in Table 1, ResNet50). We verify the idea by plotting the training and validation curves of a tiny network (MobileNetV2-Tiny) and a large network (ResNet50) in Figure 4, with and without applying NetAug. Applying NetAug improves the training and validation accuracy of MobileNetV2-Tiny, demonstrating that NetAug effectively reduces the under-fitting issue. For ResNet50, NetAug improves the training accuracy while lowers the validation accuracy, showing signs of overfitting.

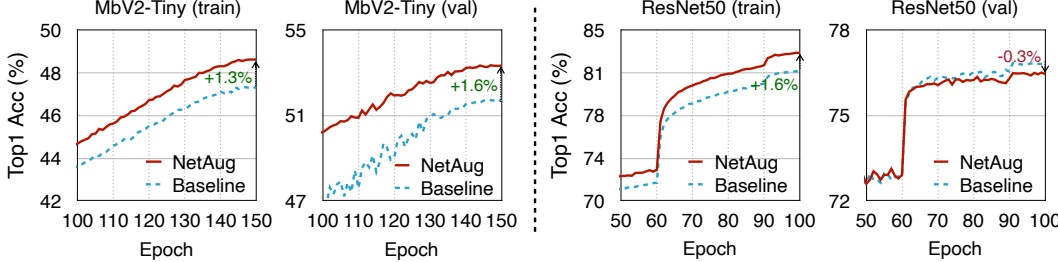

Figure 4: Learning curves on ImageNet. *Left:* NetAug alleviates the under-fitting issue of tiny neural networks (e.g., MobileNetV2-Tiny), leading to higher training and validation accuracy. *Right:* Larger networks like ResNet50 does not suffer from under-fitting; applying NetAug will exacerbate over-fitting (higher training accuracy, lower validation accuracy).

| Method | | ImageNet Top1 (%) | Fine-grained Classificaition: Top1 (%) | | | | | Det: AP50 (%) | |
|---|---|---|---|---|---|---|---|---|---|
| | | | Food101 | Flowers102 | Cars | Cub200 | Pets | VOC | COCO |
| MbV2 w0.35 r160 | Baseline (150) | 56.3 | 76.4 | 90.8 | 76.9 | 69.0 | 81.9 | 60.4 | 24.7 |
| | Baseline (300) | 57.0 | 76.5 | 90.3 | 75.8 | **69.6** | 81.7 | 60.8 | - |
| | Baseline (600) | 57.5 | 76.8 | 89.7 | 74.7 | 69.5 | 81.7 | 61.3 | - |
| | KD | 57.0 | 76.4 | 91.6 | 78.4 | 68.4 | 80.3 | 60.0 | 24.3 |
| | NetAug | 57.8 | 77.4 | 92.4 | 79.8 | 68.8 | **82.3** | **62.4** | **25.4** |
| | NetAug+KD | **59.2** | **77.5** | **92.9** | **80.4** | 68.5 | 82.2 | 62.1 | **25.4** |
| MbV3 w0.35 r160 | Baseline (150) | 58.1 | 76.6 | 92.1 | 75.8 | 69.3 | 83.1 | 63.6 | 29.2 |
| | KD | 59.8 | 76.9 | 92.6 | 77.2 | 69.2 | 82.9 | 63.4 | 28.9 |
| | NetAug | 60.3 | **78.3** | **93.0** | **78.9** | 70.4 | **84.6** | 65.3 | 30.7 |
| | NetAug+KD | **61.5** | 77.1 | **93.0** | 78.6 | **70.5** | 84.5 | **66.0** | **30.8** |

Table 4: Transfer learning results of MobileNetV2 (w0.35, r160) and MobileNetV3 (w0.35, r160) with different pre-training methods. In most cases, models pre-trained with NetAug provide the best transfer learning performance on fine-grained classification and object detection. Results that are worse than the 'Baseline (150)' are in red, and results that are better than the 'Baseline (150)' are in green. Best results are highlighted in bold.

## 4.3 RESULTS ON TRANSFER LEARNING

Models pre-trained on ImageNet are usually used for initialization in downstream tasks such as fine-grained image classification (Cui et al., 2018; Kornblith et al., 2019; Cai et al., 2020b) and object detection (Everingham et al., 2010; Lin et al., 2014). In this subsection, we study whether NetAug can benefit these downstream tasks using five fine-grained image classification datasets and two object detection datasets. The input image size is 160 for fine-grained image classification and 416 for object detection.

The transfer learning results of MobileNetV2 w0.35 and MobileNetV3 w0.35 with different pre-trained weights are summarized in Table 4. We find that a higher accuracy on the pre-training dataset (i.e., ImageNet in our case) does not always lead to higher performances on downstream tasks. For example, though adding KD improves the ImageNet accuracy of MobileNetV2 w0.35 and MobileNetV3 w0.35, using weights pre-trained with KD hurts the performances on two fine-grained classification datasets (Cub200 and Pets) and all object detection datasets (Pascal VOC and COCO). Similarly, training models for more epochs significantly improves the ImageNet accuracy of MobileNetV2 w0.35 but hurts the performances on three fine-grained classification datasets.

Compared to KD and training for more epochs, we find models pre-trained with NetAug achieve clearly better transfer learning performances in most cases, though their ImageNet performances are similar. It shows that encouraging the tiny models to work as a sub-model of larger models not only

| Method | | Fine-grained Classificaition: Top1 (%) | | | | |
| --- | --- | --- | --- | --- | --- | --- |
| | | Food101 | Flowers102 | Cars | Cub200 | Pets |
| MbV2 w0.35 r160 | Baseline (150) | 67.33 | 89.04 | 58.28 | 57.75 | 78.88 |
| | NetAug | 68.67 | 90.40 | 60.02 | 58.39 | 79.37 |
| MbV3 w0.35 r160 | Baseline (150) | 67.53 | 89.19 | 51.18 | 57.99 | 80.32 |
| | NetAug | 69.74 | 90.73 | 56.21 | 58.94 | 82.04 |

Table 5: NetAug also benefits the tiny transfer learning (Cai et al., 2020b) setting where pre-trained weights are frozen to reduce training memory footprint.

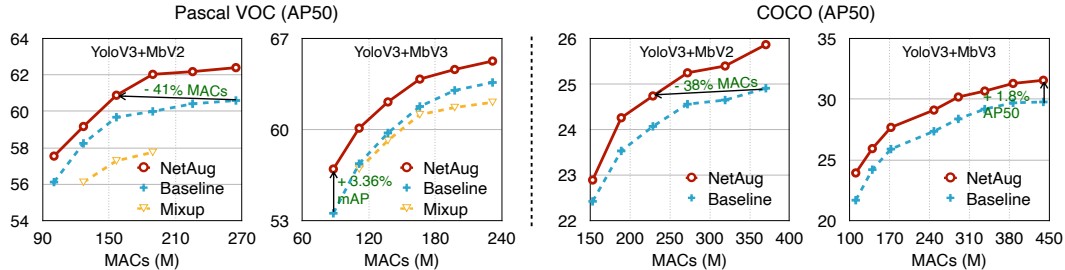

Figure 5: On Pascal VOC and COCO, models pre-trained with NetAug achieve a better performance-efficiency trade-off. Similar to ImageNet classification, adding detection mixup (Zhang et al., 2019b) that is effective for large neural networks causes performance drop for tiny neural networks.

improves the ImageNet accuracy of the models but also improves the quality of learned representation. In addition to the normal transfer learning setting, we also test the effectiveness of NetAug under the tiny transfer learning (Cai et al., 2020b) setting where the pre-trained weights are frozen while only updating the biases and additional lite residual modules to reduce the training memory footprint. As shown in Table 5, NetAug can also benefit tiny transfer learning, providing consistently accuracy improvements over the baseline.

Apart from improving the performance, NetAug can also be applied for better inference efficiency. Figure 5 demonstrates the results of YoloV3+MobileNetV2 w0.35 and YoloV3+MobileNetV3 w0.35 under different input resolutions. Achieving similar AP50, NetAug requires a smaller input resolution than the baseline, leading to 41% MACs reduction on Pascal VOC and 38% MACs reduction on COCO. Meanwhile, a smaller input resolution also reduces the inference memory footprint (Lin et al., 2020), which is also critical for running tiny deep learning models on memory-constrained devices.

Additionally, we experiment with detection mixup (Zhang et al., 2019b) on Pascal VOC. Similar to ImageNet experiments, adding detection mixup provides worse mAP than the baseline, especially on YoloV3+MobileNetV2 w0.35. It shows that the under-fitting issue of tiny neural networks also exists beyond image classification.

## 5 CONCLUSION

We propose Network Augmentation (NetAug) for improving the performance of tiny neural networks, which suffer from limited model capacity. Unlike regularization methods that aim to address the over-fitting issue for large neural networks, NetAug tackles the under-fitting problem of tiny neural networks. This is achieved by putting the target tiny neural network into larger neural networks to get auxiliary supervision during training. Extensive experiments on image classification and object detection consistently demonstrate the effectiveness of NetAug on tiny neural networks.

## ACKNOWLEDGMENTS

We thank National Science Foundation, MIT-IBM Watson AI Lab, Hyundai, Ford, Intel and Amazon for supporting this research.

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

## A    ABLATION STUDY ON REGULARIZATION METHODS

| Model (MobileNetV2-Tiny) | ImageNet Top1 Acc | ΔAcc |
|---|---|---|
| Baseline | 52.4% | - |
| Mixup ($\alpha$=0.1) (Zhang et al., 2018a) | 52.2% | -0.2% |
| Mixup ($\alpha$=0.2) (Zhang et al., 2018a) | 51.7% | -0.7% |
| DropBlock (kp=0.95, block size=5) (Ghiasi et al., 2018) | 50.6% | -1.8% |
| DropBlock (kp=0.9, block size=5) (Ghiasi et al., 2018) | 48.7% | -3.7% |
| RandAugment (N=1, M=9) (Cubuk et al., 2020) | 51.5% | -0.9% |
| RandAugment (N=2, M=9) (Cubuk et al., 2020) | 49.6% | -2.8% |
| NetAug | 53.7% | +1.3% |

Table 6: Ablation study on regularization methods. All models are trained for 300 epochs on ImageNet.

For DropBlock, we adopted the implementation from `https://github.com/miguelvr/dropblock`. Additionally, block size=7 is not applicable on MobileNetV2-Tiny, because the feature map size at the last stage of MobileNetV2-Tiny is 5. For RandAugment, we adopted the implementation from `https://github.com/rwightman/pytorch-image-models`.

As shown in Table 6, increasing the regularization strength results in a higher accuracy loss for the tiny neural network. Removing regularization provides a better result.

## B    ABLATION STUDY ON TRAINING SETTINGS

In addition to NetAug, there are several simple techniques that potentially can alleviate the under-fitting issue of tiny neural networks, including using a smaller weight decay, using weaker data augmentation, and using a smaller batch size. However, as shown in Table 7, they are not effective compared with NetAug on ImageNet.

| Model (MobileNetV2-Tiny) | ImageNet Top1 Acc |
|---|---|
| Baseline | 51.7% |
| Smaller Weight Decay (1e-5) | 51.6% |
| Smaller Weight Decay (1e-6) | 51.3% |
| Weaker Data Augmentation (No Aug) | 50.8% |
| Smaller Batch Size (1024) | 51.5% |
| Smaller Batch Size (256) | 51.7% |
| NetAug | 53.3% |

Table 7: Ablation study on training settings. All models are trained for 150 epochs on ImageNet.

