# OpenReview forum: "Network Augmentation for Tiny Deep Learning"
_ICLR.cc/2022/Conference — ICLR 2022 Poster_

### Official Review · Reviewer_ME7S · 2021-11-01

**Correctness:** 3
**Technical Novelty And Significance:** 2
**Empirical Novelty And Significance:** 3
**Recommendation:** 6
**Confidence:** 3

**Main Review:**

### Strengths
S1. This paper has a clear motivation. \
S2. This paper is easy-to-read and well-organized. \
S3. The proposed method is simple, so it has broad applicability. \
S4. The proposed method shows consistent improvements in a variety of settings and high compatibility with other frameworks, including knowledge distillation and network pruning.

### Weaknesses
W1. How does the proposed method solve the under-fitting issue? \
I agree with the under-fitting issue in training tiny neural networks. However, even though augmented networks are larger than the target tiny networks, the capacity of the target network is fixed (i.e., constant). Hence, the under-fitting issue of the target network still exists and might remain unsolved. I wonder why and how the proposed method can find a better optimum.

W2. Missing baselines that can avoid under-fitting. \
There are several simple techniques to avoid under-fitting. For example, one can use a smaller weight decay (e.g., 1e-5, 1e-6) or weaker data augmentations. Also, using a small batch size can be another option since randomness may prevent stuck in the local optimum.

### Minor comments
- The citation formats in tables are not consistent with that in the main body.


**Summary Of The Paper:**

When training tiny neural networks, a major issue is under-fitting. This paper proposes Network Augmentation (NetAug), which randomly augments (enlarges) neural networks during training to solve the under-fitting issue. NetAug shows consistent improvements when training tiny neural networks in various settings, including ImageNet classification, fine-grained classification, and object detection.


**Summary Of The Review:**

I think this paper's strengths are clear, but some important parts (explanation and baselines) are missing. Hence, my initial rating is weak reject. I will raise my rating if all the issues are resolved with a strong rebuttal in the discussion period.

---

> ### Author Response · Authors · 2021-11-22
> **Response to Reviewer ME7S**
>
> Thank you so much for the insightful and valuable comments! They are very helpful for further improving the clarity and quality of our paper. We have updated our manuscript to address all of your concerns.
>
> **[1. How does the proposed method solve the under-fitting issue?]**
>
> Thanks for raising this important question. The final performance of a tiny neural network depends not only on its capacity but also on the optimization process. Our method does not change the model capacity. Instead, it provides **auxiliary supervision** for the target model during the training process (Figure 2), which **alleviates the optimization difficulty** and improves the performance.
>
> From the perspective of local optimums, our method can be viewed as adding a prior about good local optimums into the training process: *a good local optimum should produce useful representations as a sub-model for larger neural networks in addition to functioning independently*. Adding this prior eliminates local optimums that do not satisfy this property, shrinking the space of local optimums. We conjecture it is easier for tiny neural networks to reach a good local optimum in this sub-space than in the original space. Our paper empirically shows that it could lead to better performances without changing the model architecture. We believe theoretical analysis is promising and important, and we leave it to future work.
>
> **[2. Missing baselines]**
>
> Thanks for your suggestions. We compared with the suggested baselines (smaller weight decay, weaker data augmentation, smaller batch size) to further strengthen our paper. Baseline experiments are conducted on ImageNet using MobileNetV2-Tiny (Table 9). **Our method clearly outperforms these baselines on ImageNet.** The results are summarized below:
>
> - Weight decay (wd). As shown in the following table, reducing the weight decay from 4e-5 to 1e-5/1e-6 hurts the accuracy on ImageNet. The original weight decay 4e-5 is already small enough for the tiny neural network; further decreasing the weight decay does not help. In contrast, NetAug brings significant improvement.
>
> | Method | NetAug (ours) | Baseline (wd = 4e-5) | Baseline (wd = 1e-5) | Baseline (wd = 1e-6) |
> |:-----------------:|:----------:|:----------:|:----------:|:----------:|
> | ImageNet Top1 | 53.0% | 51.2% | 51.0% | 50.8% |
>
> - Data augmentation (DA). In fact, we already used a very weak data augmentation that only includes basic transformations (random resized crop, random horizontal flip) in our experiments. Removing these basic transformations hurts the accuracy on ImageNet.
>
> | Method | NetAug (ours) | Baseline (Basic DA) | Baseline (No DA) |
> |:-----------------:|:----------:|:----------:|:----------:|
> | ImageNet Top1 | 53.0% | 51.2% | 50.8% |
>
> - Batch size (bs). We find the ImageNet accuracy of the baseline does not change a lot (only 0.1% difference) under different batch sizes (from 256 to 2048), similar to [1]. It shows that the randomness from using a small batch size cannot effectively solve the under-fitting issue of tiny neural networks on a large-scale dataset like ImageNet. In contrast, our method that provides explicit auxiliary guidance for tiny neural networks is more effective.
>
> | Method | NetAug (ours) | Baseline (bs = 2048) | Baseline (bs = 1024) | Baseline (bs = 512) | Baseline (bs = 256) |
> |:-----------------:|:----------:|:----------:|:----------:|:----------:|:----------:|
> | ImageNet Top1 | 53.0% | 51.2% | 51.2% | 51.1% | 51.3% |
>
>
> **[3. Citation formats in tables]**
>
> Thank you for the detailed comment. We have fixed the format issue in our revised manuscript.
>
> **We hope our response has resolved all of your concerns, and turns your assessment to the positive side.** *Please do not hesitate to contact us if there are other clarifications or experiments we can offer.*
>
> Thank you for your time!
>
> Best wishes,
> Authors
>
> **References**
>
> [1] Goyal, Priya, et al. "Accurate, large minibatch sgd: Training imagenet in 1 hour." arXiv preprint arXiv:1706.02677 (2017).

---

> > ### Comment · Reviewer_ME7S · 2021-11-23
> > **Thanks for the detailed response. I'll update my score.**
> >
> > Thank you for your efforts in this response. The detailed explanations and additional experiments addressed my concerns. I hope these discussions will be incorporated into the final draft. I'll update my score.

---

> > > ### Author Response · Authors · 2021-11-25
> > > **Response to Reviewer ME7S**
> > >
> > > Dear Reviewer ME7S,
> > >
> > > Many thanks for all the helpful comments. We are glad that our response has addressed your concerns. We promise these discussions will be incorporated into the final draft of our paper.
> > >
> > > Thank you again for your time and support!
> > >
> > > Best wishes,
> > > Authors

---

### Official Review · Reviewer_PXn6 · 2021-11-01

**Correctness:** 3
**Technical Novelty And Significance:** 3
**Empirical Novelty And Significance:** Not applicable
**Recommendation:** 6
**Confidence:** 5

**Main Review:**

Generally, I like the idea proposed in this paper. It is clear, somewhat interesting and looks reasonable. While, it is essentially similar to the one-shot NAS with weight sharing method, even the authors emphasized that they are different.

In the weight sharing training, the sub-network can also be augmented by auxiliary supervision across different iterations, it seems this proposed method just aggregates the gradients from auxiliary supervisions (from larger models) in a single iteration training, more like a special case of the former.

**Some of my other concerns are as follows:**

***About novelty:***

As I mentioned above, the overall idea and implementation strategy are both similar to the weight sharing mechanism.

Also, only considering width dimension without depth in this paper is too limited for the method to be deployed on real-world applications.

***About experiments:***

In the paper, the authors claimed MobileNetV2-Tiny is a tiny model so the method is effective and ResNet-50 is a large one so the method is harmful, this may be true from the perspective when compared to some compact networks. But if we consider efficientnet_l2, vit_large, GPT3, etc., ResNet-50 is also a *tiny* model in this case, it seems the proposed method only works well on the extremely low-level performance families, basically, these extremely tiny models may not have fully converged so they are relatively easy for improvement. I suggest the authors can try to train the tiny models with more epochs like 600 and better optimizers like AdamW to make sure the model has completely converged (a strong baseline), then check whether the proposed method is still effective in this case.

***About KD:***

First, I personally don’t agree with the claims in Sec. 2 of the paper that “Recent studies (Furlanello et al., 2018; Cho & Hariharan, 2019; Yuan et al., 2020) show that the teacher model does not necessarily need to be larger than the student model in KD.” These papers claimed the teacher scale is not necessary but there are also a large number of literatures stating that larger models can provide more precise supervision and are crucial for better distillation, such as: A good teacher is patient and consistent [1], Is label smoothing truly incompatible with knowledge distillation [2] ReLabel ImageNet [3], etc.

Basically, this is not a well-explored problem and in different scenarios the requirements for the teachers in KD are definitely different, so it seems the claims in this paper are too arbitrary and not well motivated. I just went through all the mentioned related papers, it seems the performance of the models in these papers is still at a very low level and they mainly tested on small datasets like CIFAR, which means the models may not yet fully converge. I don’t think the derived conclusions from these papers are more solid than [1] which has true state-of-the-art accuracy on the large-scale ImageNet.

[1] Lucas Beyer, Xiaohua Zhai, Am ́elie Royer, Larisa Markeeva, Rohan Anil, and Alexander Kolesnikov. Knowledge distillation: A good teacher is patient and consistent. arXiv preprintarXiv: 2106.05237, 2021.

[2] Zhiqiang Shen, Zechun Liu, Dejia Xu, Zitian Chen, Kwang-Ting  Cheng, and Marios Savvides. Is label smoothing truly incompatible with knowledge distillation: An empirical study. In ICLR, 2021.

[3] Sangdoo Yun, Seong Joon Oh,  Byeongho Heo,  DongyoonHan, Junsuk Choe, and Sanghyuk  Chun. Re-labeling imagenet: from single to multi-labels, from global to localized labels. In CVPR, 2021.

Furthermore, KD seems not related to the proposed method, I’m a little bit confused why the authors spend a lot of space to introduce KD in the related work and further involve it into all the experiments. To my perspective, some statements are even not accurate and also not yet verified.

KD brings the significant contribution of improvement, but it will also involve much uncertainty and vagueness of the improvement. I think it's better for the authors to focus on the proposed method itself in the experiments. Considering the not very accurate statements for the KD introduction, I’m concerned about the solidness of the experiments in this paper.


**Summary Of The Paper:**

This paper proposed to train and augment a tiny network by incorporating it into the larger networks with weight sharing training mechanism. The tiny neural network is learned with the auxiliary/additional supervisions from the larger models that wrap it. With this training strategy, the tiny model can perform better than the conventional training scheme on ImageNet and several downstream tasks.

**Summary Of The Review:**

I think this is an interesting paper, but I still have a few concerns stated above, especially on the experimental part and some statements on knowledge distillation.

---

> ### Author Response · Authors · 2021-11-22
> **Response to Reviewer PXn6 (2/2)**
>
> **[3. Novelty compared to one-shot NAS]**
>
> We would like to highlight several key differences between our work and one-shot NAS:
>
> - NetAug is solving a fundamentally different problem from one-shot NAS. NetAug improves the accuracy of a tiny neural network, while one-shot NAS searches for a model architecture under efficiency constraints. Improving the accuracy of tiny model is highly important in practice, as it can benefit many real-world deep learning applications on edge devices where neural networks must be extremely small to run efficiently. However, as pointed out by **Reviewer co48**, "there are relatively fewer works focusing on the tiny network training". NetAug is novel and timely.
> - Though one-shot NAS also uses the weight sharing strategy, it is mainly used to improve the training efficiency of the super-net.  Whether and how weight sharing can help improve the performance of a given tiny neural network is an open question. Our method provides a completely new insight towards answering this question: *encouraging the tiny model to work as a sub-model of larger neural networks provides useful auxiliary supervision for the tiny model*.
> - Our augmentation strategy is also different from one-shot NAS. In our method, auxiliary supervision only comes from larger neural networks that contain the target model. In one-shot NAS, a sub-network gets auxiliary supervision from both larger sub-networks and smaller sub-networks. This difference is very crucial in our case, as encouraging subsets of the neural network to produce predictions is a kind of dropout, which harms the model capacity of tiny neural networks.
>
> **[4. Only considering the width dimension is limited.]**
>
> We would like to clarify that
> - Our method can also support the depth dimension. We prioritize exploring the width dimension in our practical implementations because augmenting the width incurs less training overhead on GPU than augmenting the depth. We think exploring whether augmenting the depth dimension can improve the performance is an interesting question and potentially can further improve the performances of our method. We leave it to future work.
>
> **We hope our response has resolved all of your concerns, and turns your assessment to the positive side.** *Please do not hesitate to contact us if there are other clarifications or experiments we can offer.*
>
> Thank you for your time!
>
> Best wishes,
> Authors

---

> > ### Comment · Reviewer_PXn6 · 2021-11-23
> > **I appreciate authors' responses**
> >
> > I appreciate the authors' responses, while I still have the following concerns that the authors did not address in their rebuttal:
> >
> > **the relationship to weight sharing mechanism** The proposed method is essentially a weight sharing strategy, which has been heavily studied in previous literature and I feel the proposed approach has very limited novelty.
> >
> > **why the proposed method is not effective on ResNet-50** The authors did not mention this in their rebuttal. In their manuscript, they claimed that ResNet-50 is too **large** (over-fitting issue) so their method is not suitable or even harmful, this is not convincing and somewhat unreasonable as the recent study has shown that well-designed training settings and hyper-parameters will improve ResNet-50 from 76%+ to 80.4% [1], so I think it is not an **over-fitting** problem and basically ResNet-50 is still under-fitting. The concepts of **large** or **tiny** depend on how we define them. I feel the proposed method may be only helpful in an extremely low-level accuracy family, which I'm not sure is **truly helpful** or just because the model is not well-trained.
> >
> > [1] Wightman, Ross, Hugo Touvron, and Hervé Jégou. "ResNet strikes back: An improved training procedure in timm." arXiv preprint arXiv:2110.00476 (2021).

---

> > > ### Author Response · Authors · 2021-11-25
> > > **Thank you for the feedback. Our clarifications (4/4)**
> > >
> > > > #### 4. The proposed method is essentially a weight sharing strategy, which has been heavily studied in previous literature and I feel the proposed approach has very limited novelty.
> > >
> > > We respectfully disagree. What we really focus on advocating in this work is a novel training method along with the new insight for improving the performances of tiny neural networks. Please check **Reviewer co48**'s nice summarization of our contribution: "While there are lots of works studying how to improve the accuracy of large models, there are relatively fewer works focusing on the tiny network training. This work demonstrates that the tiny models suffer from under-fitting rather than over-fitting therefore requires different training strategies. The proposed method, NetAug, is effective and simple to implement. It also works well with other techniques such as knowledge distillation and pruning.".
> > >
> > > Additionally, our task, key insight, and training process are fundamentally different from previous papers (e.g., one-shot NAS) that use the weight sharing strategy, as discussed in our previous response ([3. Novelty compared to one-shot NAS]).
> > >
> > > We summarize our key contributions below to make it more clear:
> > > - We introduce a novel training method for improving the performance of tiny neural networks. Its underlying mechanism is different from existing methods such as KD and pruning. It can be combined with KD and pruning to further boost the performances of tiny neural networks.
> > > - We provide a new insight for tiny neural network training: encouraging the tiny neural network to work as a sub-model of larger neural networks can alleviate the optimization difficulty of tiny neural networks and improve their performances. It can be viewed as adding a prior about good local optimums into the training process: *a good local optimum should produce useful representations as a sub-model for larger neural networks in addition to functioning independently*. In contrast, applying regularization techniques designed to alleviate the over-fitting issue of large models hurts the accuracy of tiny neural networks.
> > > - Extensive experiments on diverse settings consistently justify the effectiveness of our method, which can benefit lots of tiny deep learning applications on edge devices.
> > >
> > > **We hope our response has resolved all of your concerns, and turns your assessment to the positive side.** *Please do not hesitate to contact us if there are other clarifications or experiments we can offer.*
> > >
> > > Thank you for your time!
> > >
> > > Best wishes,
> > > Authors

---

> > > > ### Comment · Reviewer_PXn6 · 2021-11-25
> > > > **Thanks for the more clarifications**
> > > >
> > > > Hi Authors,
> > > >
> > > > I think this manuscript definitely has merits and is worth obtaining more attention in the relevant community (as I said, I like the idea proposed in this paper), so I am willing to upgrade my rating. However, I feel the responses from the authors may be a little bit too arrogant. I hope the authors can update their revision accordingly as they promised.

---

> > > > > ### Author Response · Authors · 2021-11-26
> > > > > **Response to Reviewer PXn6**
> > > > >
> > > > > Dear Reviewer PXn6,
> > > > >
> > > > > Happy Thanksgiving.
> > > > >
> > > > > Many thanks for appreciating the merits of our work! We sincerely apologize if the words look arrogant to you. We promise we will further revise our manuscript to make our paper more clear following your constructive suggestions. All stated changes will be incorporated into the final version of our paper.
> > > > >
> > > > > Thank you again for your time and support!
> > > > >
> > > > > Best wishes,
> > > > > Authors

---

> > > ### Author Response · Authors · 2021-11-25
> > > **Thank you for the feedback. Our clarifications (3/4)**
> > >
> > > > #### 3. I feel the proposed method may be only helpful in an extremely low-level accuracy family, which I'm not sure is truly helpful or just because the model is not well-trained.
> > >
> > > First, we are targeting a very important real-world problem, improving the performance of neural networks for tiny edge devices. Due to the tight hardware constraint, models under this scenario have relatively lower accuracy compared with cloud models. There is a huge need to improve the accuracy of tiny neural network models.
> > >
> > > Second, **our baseline models are already well-trained**: the public ImageNet top1 accuracy of MobileNetV2 (w1.0, r160) is 68.8% [1] while our baseline accuracy for MobileNetV2 (w1.0, r160) is 69.6%. Compared with this much stronger baseline, our method still provides a significant accuracy improvement (+0.9%) with little training overhead. **It is not fair to criticize our experiments just because we are not targeting large neural networks. Our focus is tiny deep learning.**
> > >
> > > Third, we want to point out that the effectiveness of our method has been justified in **extensive experiments** under **different settings** (with/without KD, with/without pruning), **different datasets** (ImageNet, ImageNet-21K-P), and **different tasks** (classification, object detection). We have also compared our method with extra baselines suggested by you and Reviewer ME7S. All these experiments consistently demonstrate the benefits of our method.
> > >
> > > Both **Reviewer co48** and **Reviewer ME7S** find our experiment results convincing and solid: "Extensive experiments on ImageNet, Pascal VOC and several down-stream tasks show the superiority of the proposed method" by **Reviewer co48**; "consistent improvements in a variety of settings and high compatibility with other frameworks, including knowledge distillation and network pruning" by **Reviewer ME7S**.
> > >
> > > **References**
> > >
> > > [1] https://github.com/tensorflow/models/tree/master/research/slim/nets/mobilenet

---

> > > ### Author Response · Authors · 2021-11-25
> > > **Thank you for the feedback. Our clarifications (2/4)**
> > >
> > > > #### 2. ResNet-50 is still under-fitting.
> > >
> > > We respectfully disagree.
> > > - First, if a model over-fits the training data, alleviating its over-fitting issue can also improve the accuracy on the validation set. **Therefore, solely based on the improved validation accuracy, we cannot conclude whether the model over-fit or under-fit the training data.**
> > > - Second, abundant evidence in the literature [2,3,4,5] shows that adding strong regularization techniques that insert noise to the network or the dataset can improve the ImageNet validation accuracy of ResNet-50 while lowering its training accuracy. It is a typical phenomenon when the model is over-fitting (rather than under-fitting). **It is a common belief that ResNet-50 tends to over-fit the training data on ImageNet, thus applying regularization techniques can improve its validation accuracy [2,3,4,5]. In contrast, we never see any evidence or work showing that ResNet-50 under-fits the training data on ImageNet.**
> > > - Third, we want to point out that the high ImageNet validation accuracy of ResNet-50 reported in [1] largely relies on these regularization techniques, including RandAugment [2], Mixup [4], CutMix [3], Stochastic-Depth [6], etc. Therefore, [1]'s results actually positively support that ResNet-50 suffers from the over-fitting issue on ImageNet.
> > >
> > > ***All these facts and results show that ResNet-50 suffers from the over-fitting issue on ImageNet (i.e., ResNet-50 tends to over-fit the ImageNet training data). We sincerely hope the AC and other reviewers can provide their valuable opinions to help resolve this question.***
> > >
> > > Since our method is not designed to prevent the model from over-fitting the training data (Figure 4), it is not surprising that our method does not improve the performance of ResNet-50. However, it does not hurt our method's value for tiny deep learning, which is the focus of our work.
> > >
> > > **References**
> > >
> > > [1] Wightman, Ross, Hugo Touvron, and Hervé Jégou. "ResNet strikes back: An improved training procedure in timm." arXiv preprint arXiv:2110.00476 (2021).
> > >
> > > [2] Cubuk, Ekin Dogus, et al. "RandAugment: Practical Automated Data Augmentation with a Reduced Search Space." Advances in Neural Information Processing Systems 33 (2020).
> > >
> > > [3] Yun, Sangdoo, et al. "Cutmix: Regularization strategy to train strong classifiers with localizable features." Proceedings of the IEEE/CVF International Conference on Computer Vision. 2019.
> > >
> > > [4] Zhang, Hongyi, et al. "mixup: Beyond Empirical Risk Minimization." International Conference on Learning Representations. 2018.
> > >
> > > [5] Ghiasi, Golnaz, Tsung-Yi Lin, and Quoc V. Le. "DropBlock: A regularization method for convolutional networks." Advances in Neural Information Processing Systems 31 (2018): 10727-10737.
> > >
> > > [6] Huang, Gao, et al. "Deep networks with stochastic depth." European conference on computer vision. Springer, Cham, 2016.

---

> > > ### Author Response · Authors · 2021-11-25
> > > **Thank you for the feedback. Our clarifications (1/4)**
> > >
> > > Thanks for your additional feedback.
> > >
> > > > #### 1. The concepts of large or tiny depend on how we define them.
> > >
> > > We politely yet firmly point out that your comments misread the problem setting of our paper.
> > > - **Tiny deep learning is an important research direction.** It targets on-device machine learning with resource-constrained edge devices (e.g., billions of IoT devices [1]). There is a rich research community [2,3,4,5] studying this problem, and it has many real-world applications, such as smart home, smart retail, smart manufacturing, etc. We are solving a very important challenge that has real-world impacts.
> > > - The size of a tiny neural network is determined by the actual hardware constraint, it's not a subjective concept. For instance, a microcontroller has only 1MB of Flash, therefore a tiny model should be smaller than 1MB. Our tiny model design is targeting real-world settings.
> > > - Though ResNet-50 can be somehow defined as a "tiny" model compared with ViT_large, GPT3, it is too large to be deployed on IoT devices. Thus, it is not a tiny neural network in the context of our paper.
> > >
> > > **References**
> > >
> > > [1] https://www.statista.com/statistics/471264/iot-number-of-connected-devices-worldwide/
> > >
> > > [2] https://www.tinyml.org/
> > >
> > > [3] Lin, Ji, et al. “MCUNet: Tiny Deep Learning on IoT Devices.” Advances in Neural Information Processing Systems 33 (2020): 11711-11722.
> > >
> > > [4] Saha, Oindrila, et al. “RNNPool: Efficient Non-linear Pooling for RAM Constrained Inference.” Advances in Neural Information Processing Systems 33 (2020).
> > >
> > > [5] Banbury, Colby, et al. “Micronets: Neural network architectures for deploying tinyml applications on commodity microcontrollers.” Proceedings of Machine Learning and Systems 3 (2021).

---

> > > > ### Comment · Reviewer_PXn6 · 2021-11-25
> > > > **misunderstandings on my concerns**
> > > >
> > > > I think the authors misunderstood my concerns. I'm not talking about the concept of "tiny" or "large" in real-world scenarios. I also agree that network training design on resource-constrained edge devices is important. My concern is based on the paper's core claim: **large models suffer from over-fitting like ResNet-50 and tiny models tend to under-fit like MobileNetV2-Tiny**. So I would like to know how can we define tiny or large from this perspective?
> > > >
> > > > I also would like to clarify that it may be not true "a microcontroller has only 1MB of Flash, therefore a tiny model should be smaller than 1MB" so models>1MB will be overfitting and <1MB will be under-fitting. I think they are not directly related, and "a microcontroller with1MB Flash" only reflects the current level of technology.

---

> > > > > ### Author Response · Authors · 2021-11-26
> > > > > **Response to Reviewer PXn6**
> > > > >
> > > > > Dear Reviewer PXn6,
> > > > >
> > > > > Thank you for clarifying your concern. Our previous discussions are from the perspective of hardware. From the perspective of over-fitting and under-fitting, we think the definition of large or tiny depends on the dataset/task. For example, in the claim "large models suffer from over-fitting like ResNet-50 and tiny models tend to under-fit like MobileNetV2-Tiny", we assume the target task is ImageNet classification. When targeting a much larger dataset (e.g., JFT-300M), it is possible that ResNet-50 also does not have sufficient capacity, thereby suffering from the under-fitting issue. We will make this more clear in our revised manuscript.
> > > > >
> > > > > Best wishes,
> > > > > Authors

---

> ### Author Response · Authors · 2021-11-22
> **Response to Reviewer PXn6 (1/2)**
>
> Thank you so much for the insightful and valuable comments! They are very helpful for further improving our paper. We have updated our paper following your suggestions.
>
>
> **[1. Remove discussions about KD that are unrelated to the proposed method and focus on the proposed method itself in the experiments.]**
>
> Thanks for the great suggestions! We agree KD is not related to our method. In our revised manuscript, we have removed these unrelated discussions. In addition, as you suggested, we have removed KD for both the baseline and our method. **Our method still shows consistent improvements on various neural network architectures without KD.** (see table below)
>
> | Model | MobileNetV2-Tiny (r144) | MobileNetV2 (r160, w0.35) | ProxylessNAS Mobile (r160, w0.35) | MCUNet (r176) | MobileNetV3 (r160, w0.35) |
> |:-----------------:|:----------:|:----------:|:----------:|:----------:|:----------:|
> | Params | 0.75M | 1.66M | 1.78M | 0.74M | 2.2M |
> | MACs | 23.5M | 30.9M | 35.7M | 81.8M | 19.6M |
> | Baseline | 51.2% | 56.3% | 59.1% | 61.4% | 58.1% |
> | NetAug (ours) | **53.0%** | **57.9%** | **60.5%** | **62.5%** | **60.4%** |
>
> **[2. Stronger baseline with longer training epochs and advanced optimizer]**
>
> Thank you for suggesting a new baseline. We have conducted experiments for this baseline on ImageNet (SGD 300/600 epochs, AdamW 600 epochs) and incorporated the results into our paper (Figure 3). **Our method still provides clear accuracy improvements over the baseline under all settings. With a similar accuracy, our method requires much fewer training epochs than the baseline.**
>
> | MobileNetV2-Tiny (ImageNet, top1) | 150 epochs (SGD) | 300 epochs (SGD) | 600 epochs (SGD) | 600 epochs (AdamW) |
> |:-----------------:|:----------:|:----------:|:----------:|:----------:|
> | Baseline | 51.2% | 51.8% | 52.7% | 52.5% |
> | NetAug (ours) | **53.0%** | **53.4%** | **54.0%** | - |
>
> We find that:
> - Training for more epochs improves the ImageNet top1 accuracy for both the baseline and our method.
> - **Our method still clearly outperforms the baseline when trained for more epochs.**
> - Using AdamW instead of SGD does not improve the accuracy of the baseline. Considering that SGD is more commonly used in previous papers when training CNNs, we continue using SGD in our experiments.
>
> In addition to ImageNet, we further conduct experiments on ImageNet-21K-P, a pre-processed version of ImageNet-21K introduced in [1], which provides a standard train-validation split.
>
> | MobileNetV2-Tiny (ImageNet-21K-P, top1) | 20 epochs (SGD) | 60 epochs (SGD) | 120 epochs (SGD) |
> |:-----------------:|:----------:|:----------:|:----------:|
> | Baseline | 22.4% | 23.4% | 24.1% |
> | NetAug (ours) | **24.3%** | **25.0%** | **25.4%** |
>
> We find that:
> - Consistent with observations on ImageNet, our method shows clear accuracy improvements over the baseline under all settings on ImageNet-21K-P.
> - Apart from the accuracy improvements, we want to highlight the benefits of our method for saving the training cost. In practice, training models for 600 epochs on ImageNet or 120 epochs on ImageNet-21K-P is too expensive and causes a lot of CO2 emissions. **Our method reduces the number of training epochs by 4x on ImageNet and 6x on ImageNet-21K-P while still achieving better accuracy than the baseline.**
>
> **References**
>
> [1] Ridnik, Tal, et al. "Imagenet-21k pretraining for the masses." NeurIPS Datasets and Benchmarks Track (2021).

---

### Official Review · Reviewer_co48 · 2021-11-02

**Correctness:** 4
**Technical Novelty And Significance:** 3
**Empirical Novelty And Significance:** 3
**Recommendation:** 8
**Confidence:** 4

**Main Review:**

[Strengths]

1) The authors show that large models benefit from data augmentation and dropout while tiny models suffer from these regularizations/augmentations.

2) Augmenting the network width to improve the trainability of tiny models is an interesting approach.

3) Extensive experiments on ImageNet, Pascal VOC and several down-stream tasks show the superiority of the proposed method.

4) Ablation studies show that the proposed method can be combined with KD and pruning to get better results.

[Weaknesses]

1) The proposed method is similar to progressive shrinking used in OFANet in principle.

2) Augmenting the width of the network in training  has been proposed before, such as "Go Wide, Then Narrow: Efficient Training of Deep Thin Networks. ICML 2020.". It would be better to discuss the key differences.


**Summary Of The Paper:**

This work proposes a novel method to train tiny networks. The proposed method first augments the width of the tiny network to make the tiny network a bit larger. Then the gradients from the larger network are used as additional supervision. The authors show that the proposed method significantly improves the generalization performance of tiny models, is complementary to knowledge distillation and pruning. In addition, the authors show that data augmentation and dropout or similar regularization methods actually hurt the tiny models in training.

**Summary Of The Review:**

While there are lots of works studying how to improve the accuracy of large models, there are relatively fewer works focusing on the tiny network training. This work demonstrates that the tiny models suffer from under-fitting rather than over-fitting therefore requires different training strategies. The proposed method, NetAug, is effective and simple to implement. It also works well with other techniques such as knowledge distillation and pruning. There are existing works sharing a similar idea of NetAug for large model training, which slightly hurts the novelty of this work.

---

> ### Author Response · Authors · 2021-11-22
> **Response to Reviewer co48**
>
> Thank you so much for the positive rating and insightful comments. Your valuable suggestions are very helpful for further strengthening our paper. We have revised our paper according to your advice.
>
> **[1. Key differences between our work and "Go Wide, Then Narrow"]**
>
> Thank you for bringing "Go Wide, Then Narrow" [1] to our attention. We have added discussions about this work in our paper. We summarize the key differences between our work and [1] as follows:
> - Our work is based on the key idea of encouraging the target model to provide useful representation for wider networks to get auxiliary supervision during training. Its underlying mechanism is different from KD and network pruning. In contrast, [1]'s core idea is to employ a well-trained wider network to initialize the target model and train the target model to mimic the intermediate outputs of the wider network. It can be viewed as combining network pruning and layer-wise KD. Thus, we think our method is also orthogonal to [1] and can be combined for better performances.
>
> **[2. Compared to OFANet]**
>
> We would like to highlight several key differences between our work and OFANet [2]
> - Our work targets improving the performances of a given tiny neural networks, while OFANet targets training a super-net to support all sub-networks in the design space. Our objective and training process are different from OFANet's.
> - In OFANet, a sub-network needs to support both larger sub-networks and smaller sub-networks, while the target model in our work only needs to support larger sub-networks. This is a crucial difference in our case since encouraging subsets of the neural network to produce predictions is a kind of dropout, which is harmful to the capacity of tiny neural networks.
>
> *Please do not hesitate to contact us if there are other clarifications or experiments we can offer.*
>
> Thank you for your time!
>
> Best wishes,
> Authors
>
> **References**
>
> [1] Zhou, Denny, et al. "Go wide, then narrow: Efficient training of deep thin networks." International Conference on Machine Learning. PMLR, 2020.
>
> [2] Cai, Han, et al. "Once-for-All: Train One Network and Specialize it for Efficient Deployment." International Conference on Learning Representations. 2020.

---

> > ### Comment · Reviewer_co48 · 2021-11-23
> > **Thank you for the replies and I have no more questions.**
> >
> > The authors' feedback addressed all my concerns.

---

### Official Review · Reviewer_XSG2 · 2021-11-02

**Correctness:** 2
**Technical Novelty And Significance:** 2
**Empirical Novelty And Significance:** 2
**Recommendation:** 3
**Confidence:** 4

**Main Review:**

The paper introduced a training mechanism that includes auxiliary forward flow and supervision from the very training of the very large model into the training of the tiny nets, this training mechanism improves the classification and object detection performance of tiny nets on several large datasets. The paper does not provide evidence that if a small sub-network of the larger network which is equivalent to the tiny network in network sizes will provide the same results as shown in the paper, in other words, does the auxiliary training necessary?

**Summary Of The Paper:**

To improve the performance of compact neural networks with limited model capacity, the paper proposed Network Augmentation (NetAug) which addresses the under-fitting problem in small neural networks. This is accomplished by incorporating the target tiny neural network into bigger neural networks for additional training supervision. The paper demonstrates the effectiveness of
NetAug on improving the effectiveness and the performance of tiny models image classification and object detection on several image datasets, achieving up to 2.1% accuracy improvement on ImageNet, and 4.3% on Cars. On Pascal VOC, NetAug provides 2.96% mAP improvement with the same computational cost.

**Summary Of The Review:**

As I stated in my main review, I suspect that auxiliary training is necessary for getting the better performance of the tiny network. I suspect that an equivalent sub-network of the larger net will have similar results in comparing to the performance of the tiny network solely training on the large datasets (ImageNet, Pascal VOC etc).

---

> ### Author Response · Authors · 2021-11-22
> **Response to Reviewer XSG2**
>
> Thanks for your time and efforts. We respectfully yet firmly point out that your comments significantly misread our problem settings and results.
>
> > #### The paper does not provide evidence that if a small sub-network of the larger network which is equivalent to the tiny network in network sizes will provide the same results as shown in the paper
>
> We want to clarify that:
> - This paper targets improving the performance of a given tiny neural network, **not** searching for a new model architecture under some efficiency constraints (e.g., network size). We are very confused by your question which seems to ask for comparisons between two models with different architectures but similar model sizes.
> - The common practice ([1, 2, 3]) for justifying the effectiveness of a training technique is to use the **same neural network** and compare the performances of this neural network under different training techniques. We already provided **extensive experiments** following this common practice, **showing the effectiveness** of our method across **different datasets** (ImageNet, Pascal VOC, etc), **different neural network architectures** (MobileNetV2, MobileNetV3, ProxylessNAS, etc), and **different settings** (with/without KD, with/without pruning, etc). The practical value of our method is highly acknowledged by **Reviewer co48** as "effective and simple to implement" and "Extensive experiments on ImageNet, Pascal VOC and several down-stream tasks show the superiority of the proposed method"; by **Reviewer ME7S** as "broad applicability" and "consistent improvements in a variety of settings and high compatibility with other frameworks, including knowledge distillation and network pruning".
>
> We genuinely hope Reviewer XSG2 could kindly check our response and our updated manuscript. **We hope our response has resolved all of your concerns, and turns your assessment to the positive side.** *Please do not hesitate to contact us if there are other clarifications or experiments we can offer.*
>
> Thank you for your time, again!
>
> Best wishes,
> Authors
>
> **References**
>
> [1] Ghiasi, Golnaz, Tsung-Yi Lin, and Quoc V. Le. "DropBlock: A regularization method for convolutional networks." Advances in Neural Information Processing Systems 31 (2018): 10727-10737.
>
> [2] Zhang, Hongyi, et al. "mixup: Beyond Empirical Risk Minimization." International Conference on Learning Representations. 2018.
>
> [3] Cubuk, Ekin Dogus, et al. "RandAugment: Practical Automated Data Augmentation with a Reduced Search Space." Advances in Neural Information Processing Systems 33 (2020).

---

> ### Author Response · Authors · 2021-11-26
> **Sincerely Look Forward to Your Feedback!**
>
> Dear Reviewer XSG2,
>
> Happy Thanksgiving.
>
> Thanks again for your time. As the deadline for discussion is approaching, we really hope to have a further discussion with you to see if our response solves the concerns. We are happy to provide any additional clarifications that you may need.
>
> We sincerely appreciate it if you could reply to the most important points in our response. As we pointed out, we are targeting improving the performances of given tiny neural networks, not searching for a new model architecture under some efficiency constraints (e.g., network size). Your concern seems to refer to a non-existing issue. The merits of our work have been consistently recognized by all three other reviewers.
>
> We genuinely hope you could kindly check our response. Please do not hesitate to contact us if there are other clarifications or experiments we can offer. Thanks!
>
> Best wishes,
> Author

---

> ### Author Response · Authors · 2021-11-29
> **Additional Clarifications**
>
> Dear Reviewer XSG2,
>
> In addition to previous clarifications, we want to emphasize that
> - We already provided experimental results showing that our method is also effective when combined with network pruning (Table 4). It shows that selecting an equivalent sub-network from a pre-trained larger network and training it alone cannot achieve the same accuracy as our method.
> - We already provided experimental results showing that our method is also effective when applied to neural network architectures optimized with NAS (Table 1). Our method also consistently provides a significant accuracy improvement compared to Once-for-All, without sacrificing inference efficiency (Table 7).
>
> We are confident that our manuscript already provides strong evidence supporting the merits of our method, which has been recognized by all three other reviewers. We are also confident that our response should have cleared your concerns. Since the discussion period will end today, we genuinely hope you could kindly check our response and provide your valuable feedback. Thank you!
>
> Best,
> Authors

---

### Author Response · Authors · 2021-11-22
**General Response**

We thank all reviewers for their insightful and constructive comments. We are glad to see Reviewer **ME7S**, **PXn6** and **co48** generally appreciating our paper: interesting approach (Reviewer co48, PXn6), broad applicability (Reviewer ME7S), and consistent improvements (Reviewer co48, ME7S). We sincerely hope to have further discussions with Reviewer **ME7S** and **PXn6** to see if our responses and updates solve their concerns. We are happy to provide more information and answer any additional questions. We indeed hope to discuss further with Reviewer **XSG2**, as his/her comments seem to significantly misread our problem settings and results.

In addition to detailed responses to reviewers’ questions and concerns, we summarize the updates (highlighted with yellow in the paper) of our manuscript below:

**[1. Comparison with extra baselines]**

We have included experimental results comparing our method with baselines suggested by Reviewer PXn6 and ME7S (Figure 3 and Table 9). Our method clearly outperforms these baselines under all settings. In addition, our method requires much fewer training epochs (4x fewer on ImageNet, 6x fewer on ImageNet-21K-P) to achieve similar accuracy compared to the baseline (Figure 2), saving the training cost and reducing CO2 emissions without sacrificing accuracy.

**[2. Writing]**

- (Reviewer PXn6) We have removed discussions about KD that are unrelated to our paper. Additionally, we have updated the experiment section, removing KD for both the baseline and our method (Table 1).
- (Reviewer co48) We have added discussions about "Go Wide, Then Narrow" in the related work section.
- (Reviewer ME7S) We have fixed the citation format issue in our tables.

We thank all reviewers’ time and efforts! Please don’t hesitate to let us know of any additional comments on the manuscript or the changes.

---

### Decision · Program_Chairs · 2022-01-20

**Decision:**

Accept (Poster)

**Comment:**

This paper studies the problem of training tiny networks, by proposing a new training method called Network Augmentation (NetAug). The main challenge for training tiny networks lies in underfitting, which data augmentation and dropout etc. regularizations may suffer from for tiny networks. To overcome this hurdle, the proposed method first embeds or augments the tiny network as a subnet into a larger network, mostly by enlarging the width; then the gradients from the larger network are used as additional or auxiliary supervision. With this training strategy, the tiny model can perform better than the conventional training scheme on ImageNet and several downstream tasks. The proposed method is simple to implement and complementary with other techniques such as knowledge distillation and pruning. While there are lots of works studying how to improve the accuracy of large models, there are relatively fewer works focusing on the tiny network training. Despite that there are existing works sharing a similar idea of NetAug for large model training, which slightly hurts the novelty of this work, the majority of reviewers still like the idea and suggest to accept the paper.